# Cyanobacterial Blooms in City Parks: A Case Study Using Zebrafish Embryos for Toxicity Characterization

**DOI:** 10.3390/microorganisms12102003

**Published:** 2024-10-02

**Authors:** Bruna Vieira, João Amaral, Mário Jorge Pereira, Inês Domingues

**Affiliations:** 1Department of Biology, Campus Universitário de Santiago, 3810-193 Aveiro, Portugal; joaoamaral97@live.ua.pt; 2Department of Biology & CESAM, Campus Universitário de Santiago, 3810-193 Aveiro, Portugal; inesd@ua.pt

**Keywords:** cyanotoxins, oxidative stress, mortality, developmental delay, bloom evolution

## Abstract

Cyanobacteria are photosynthetic prokaryotes that play an important role in the ecology of aquatic ecosystems. However, they can also produce toxins with negative effects on aquatic organisms, wildlife, livestock, domestic animals, and humans. With the increasing global temperatures, urban parks, renowned for their multifaceted contributions to society, have been largely affected by blooms of toxic cyanobacteria. In this work, the toxicity of two different stages of development of a cyanobacterial bloom from a city park was assessed, evaluating mortality, hatching, development, locomotion (total distance, slow and rapid movements, and path angles) and biochemical parameters (oxidative stress, neurological damage, and tissue damage indicators) in zebrafish embryos/larvae (*Danio rerio*). Results showed significant effects for the samples with more time of evolution at the developmental level (early hatching for low concentrations (144.90 mg/L), delayed hatching for high concentrations (significant values above 325.90 mg/L), and delayed development at all concentrations), behavioral level (hypoactivity), and biochemical level (cholinesterase (ChE)) activity reduction and interference with the oxidative stress system for both stages of evolution). This work highlights the toxic potential of cyanobacterial blooms in urban environments. In a climate change context where a higher frequency of cyanobacterial proliferation is expected, this topic should be properly addressed by competent entities to avoid deleterious effects on the biodiversity of urban parks and poisoning events of wildlife, pets and people.

## 1. Introduction

Cyanobacteria are a source of a wide range of toxic secondary metabolites called “cyanotoxins” [1]. The production of these toxic compounds is often associated with the formation of cyanobacterial blooms, which appear under certain favorable conditions and are characterized by a large population of cyanobacteria that become dominant in terms of biomass and productivity [2]. Several studies show that different abiotic factors such as temperature, light intensity, pH, and nutrients influence the growth of cyanobacteria and their toxin production [3,4]. Although cyanobacterial blooms are not always accompanied by the production of toxins, their detection in aquatic systems is an increasingly recurrent phenomenon [3,4]. Climate changes are enhancing the occurrence and intensity of blooms by changing air and water temperature gradients [5,6], since warm and windless climates and low water turbulence are favorable conditions for bloom formation [2,7].

The increasing input of nutrients is also a favorable factor for the formation of blooms [8,9,10]. Phosphorus and nitrogen concentrations have a strong influence on cyanobacterial growth, potentiated through industrial, domestic, and agricultural waste discharges [11,12], which are intensified during periods of drought, when the river’s flow decreases but the input of organic and inorganic waste remains constant which, in a context of climate change, is an issue that deserves even more attention, since periods of drought have intensified. Thus, although cyanobacteria survive and proliferate in various habitats, such as salt water, desert, hot springs, and even in rocks [1], it is in eutrophic and hypertrophic lakes, rivers, and ponds around the world where most bloom formations are recorded [2]. Of the identified cyanotoxins, microcystins (MCs) are the group with the largest distribution and one of the most harmful, due to their tumor and hepatotoxic properties [13]. Although MCs are produced by other genera, such as *Anabaena* (*Dolichospermum*), *Aphanizomenon*, and *Planktothrix*, it is the genus *Microcystis* that, in addition to being the most common bloom-former, is the main producer of MCs in freshwater bodies [14]. Cyanotoxins may compromise the survival and fitness of various aquatic organisms, wild animals, as well as domestic animals and humans [14,15], leading to symptoms such as liver failure [13,16]. While most of the reports account for livestock intoxications through poisoned drinking water, other scenarios such as cyanobacterial blooms in an urban context have been raising concerns due to their proximity to people [17]. In fact, lakes in urban parks and city gardens have also been largely affected by toxic cyanobacterial blooms because they usually consist of relatively small water bodies, with limited water movement and very often receive urban-enriched effluents creating the optimal conditions for the proliferations of cyanobacteria [18,19,20]. Urban parks provide valuable ecosystem services to cities, with social, economic, and environmental benefits [21]. From an ecological perspective, urban parks are essential for the maintenance of biodiversity hotspots within the city, water purification, and regulation of the urban climate including the decrease in temperature [21,22,23]. They also provide important social and health services, since contact with nature reduces stress, promotes physical activity, and promotes sociability and a sense of belonging [24].

In this study, the zebrafish (*Danio rerio*) was chosen as a model because of its value in toxicology, due to its rapid and cost-effective assessments of chemical hazards and environmental toxicity, as well as the possibility of assessing embryonic development, such as delays and malformations, due to the embryo’s optical transparency. Moreover, the zebrafish embryo is considered an alternative model to animal experimentation and their use promoted by the European Directive 2010/63/EU. Furthermore, the zebrafish exhibits a variety of complex behaviors including social, learning, and anxiety responses, offering a powerful and versatile model system for the study of behavioral neuroscience and related fields [25,26,27,28].

The main objective of this work was to evaluate the toxicity of a cyanobacterial bloom in a lake of an urban park in the city of Aveiro (Portugal) and to understand whether different stages of evolution of a bloom represent different toxicities. Using zebrafish embryos, a multilevel assessment was performed to evaluate the mortality, hatching, development, biochemical (glutathione S-transferase (GST), catalase (CAT), glutathione peroxidase (GPx), glutathione reductase (GR), cholinesterase (ChE), and lactate dehydrogenase (LDH)) and behavioral (total distance, slow and rapid movements, and path angles) parameters.

## 2. Materials and Methods

### 2.1. Collection of Samples

The cyanobacterial material from the surface of the bloom (3 cm depth), as well as the respective water below the cyanobacteria (to a depth of 50 cm), were collected from the lake of Baixa de Santo António Park, Aveiro, Portugal (40.68620, −8.55793), in September 2022, at two different locations in the lake: at site 1, the bloom was more pasty in appearance, with brownish green coloration, further away from the water intake (presumably an older formation) and at site 2, the bloom was more diluted in appearance, with bright green coloration, located near the lake shore and the water intake (presumably a more recent formation). All the samples were collected at the same time and temperature, directly into PET bottles. The bloom samples were observed under a light microscopy (Olympus CX31, Olympus, Tokyo, Japan), and the taxonomic study was based on Komárek and coauthors’ work [29], identifying the presence of *Microcystis aeruginosa*. No cells were observed in the water column samples. Immediately after the identification, the samples were frozen at −20 °C.

### 2.2. Preparation of the Extracts

For each site, three different samples were tested: (1) extracts resulting from the lysis of the cells; (2) filtrate resulting from the cyanobacteria filtration, and (3) water column below the bloom. The samples of surface blooms were treated based on the methodology described by [30]. After the filtration (Whatman Filter paper, Grade GF/C MicroFiber Glass Filter, Binder Free) of the surface bloom, cyanobacteria samples attached to the filters were freeze-dried for at least 48 h, until dry. The resulting powder was stored in glass flasks at −20 °C until use. After freeze-drying, the dried extract from site 1 was dark green in color and dense in appearance, unlike the extract from site 2, which was lighter green and less dense. When needed, these extracts were dissolved in *D. rerio* culture water (see below) and subjected to lysis with the aid of an ultrasound tissue disrupter (Branson 250 Sonifier, parameters set to a duty cycle of 80% and an output control of 3). Sonication was performed on ice for 20s. After lysis, the cell suspension was centrifuged (10,000× *g*, 10 min, 4 °C) and the supernatant was collected for toxicological testing (cell extracts). The filtrate resulting from the filtration of the cyanobacteria samples was also sampled to assess toxicity associated to the water surrounding the cyanobacteria cells. The filtrate was stored in Falcon tubes at −20 °C. When needed, the samples were defrosted and left to reach room temperature (Figure 1).

Samples from the water column below the cyanobacterial blooms were also frozen at −20 °C. When needed, the samples were defrosted, left to reach room temperature, and tested without dilution.

### 2.3. Quantification

To determine the group and concentration of cyanotoxins in the bloom and water column samples, the Microcystin-ADDA ELISA Kit (Enzo Life Sciences, Farmingdale, NY, USA, ALX-850-319-KI01), an immunoassay for the quantitative detection of microcystins and nodularins in water samples through indirect competitive ELISA, was used according to the manufacturer’s instructions. For each bloom (site 1 and site 2), three dilutions were made: 5×, 10×, and 50×. The water column samples from both sites were analyzed with 5× and 10× dilutions. The immunoassay showed us that the samples were composed by microcystins, with no specific variant identified. However, the dilutions made were not sufficient for the environmental samples to fit the standards curve, which means that the concentration of microcystins of the samples were higher than the maximum concentration of the KIT standards (5.0 ppb).

### 2.4. Test Organisms

Adult *D. rerio* (wild type AB) were maintained in a ZebTEC (Tecniplast, Buguggiate, Italy) recirculation system, under controlled conditions, where culture water was generated by reverse osmosis, supplemented with salt (Instant Ocean Synthetic Sea Salt, Spectrum Brands, Madison, WI, USA) and automatically adjusted to a pH of 7.5 ± 0.5 and a conductivity of 794 ± 50 μS/cm. The water temperature was kept at 27.0 ± 1 °C and the dissolved oxygen level was kept at least at 95% saturation. The organisms were maintained on a 12 h:12 h (light:dark) photoperiod cycle and fed once daily with commercially available artificial diet Gemma Micro 500 (Skretting^®^, Burgos, Spain). Eggs were obtained by breeding fish in breeding tanks and after collection were gently washed with culture water. Eggs in the blastula stage [31] were selected under a stereomicroscope (Stereoscopic Zoom Microscope-SMZ 1500, Nikon, Tokyo, Japan) and used for the tests.

### 2.5. Ecotoxicity Tests

All tests were based in the Organization of Economic Co-operation and Development testing guideline 236 [32]. Embryos were exposed, for 120 h, by direct immersion in wells of 24-well polypropylene plates, one embryo per well, in 2 mL of test solution. Twenty-four embryos (replicates) were exposed per treatment. Mortality and development (development delays and malformations, edemas, and hatching) were checked daily with a stereomicroscope (Stereoscopic Zoom Microscope-SMZ 1500, Nikon) and compared with [31]. At the end of the test, zebrafish larvae were subjected to a behavioral analysis (see below). Cyanobacteria extracts of each site were tested at concentrations of 0.00, 144.90, 217.30, 325.90, 488.90, and 733.00 mg/L, which were obtained by successive dilutions of the highest concentration with fish water. Filtrate samples were tested at concentrations of 0.00, 0.07, 0.23, 0.82, 2.86, and 10.00%. The several concentrations were obtained by successive dilutions with fish water of the highest concentration solution. Samples from the water column below the cyanobacteria blooms were tested without dilution.

At 120 h post fertilization (hpf), the larvae locomotion was measured in the Zebrabox video tracking system (ZebraLab^®^ v3, Automated Behavioral Analysis) running a protocol with 6 min of acclimatization in the light, followed by 2 min of behavioral analysis in the dark. Typically, the sudden transition of light to dark conditions elicits in zebrafish larvae a burst of activity. The endpoints measured were total swimming distance (mm), distance (mm) traveled in slow movements (<7.8 mm/s) and rapid movements (≥7.8 mm/s), and the fish path angles. Angles were analyzed through the larvae’s swimming direction and the vector of turn path. The following classes of angles were considered (Appendix A): class 1: from 100° to 180°; class 2: from 40° to 100°; class 3: from 10° to 40°; and class 4: from 0° to 10°. Class 1 angles express movements with sudden changes of direction or erratic swimming, which can be interpreted as an anxiety-like behavior [33], while a normal swimming pattern will be expressed by higher percentages of class 3 and 4 angles.

### 2.6. Biochemical Analysis

A second exposure was deployed under the same conditions as those stated above, except that Petri dishes were used with 30 embryos and 40 mL of test solution in triplicate. This test aimed at collecting enough larvae for biochemical markers determinations. Solutions of cyanobacteria extract of each site were prepared at concentrations of 0.00, 46.10, 69.10, 103.70, 155.60, 233.30, and 350.00 mg/L. For the filtered samples, different dilutions were prepared, expressed as a percentage: 0.00, 0.02, 0.05, 0.13, 0.32, 0.80, and 2.00%. After 96 h of exposure, a minimum of six pools of 5 larvae and six pools of 15 larvae were sampled in microtubes and stored at −20 °C until enzymatic determinations. The 5-larvae samples were used to determine glutathione S-transferase (GST), catalase (CAT), cholinesterase (ChE), and lactate dehydrogenase (LDH). The 15-larvae samples were used to quantify glutathione peroxidase (GPx) and glutathione reductase (GR) activities. The analyses were performed in 96 wells microplates using spectrophotometric methods in a Thermo Scientific Multiskan Spectrum microplate reader. Enzymes activity was expressed by the amount of protein in the samples. Using γ-globulin as a reference, the Bradford technique [34] was used to determine the protein concentration of the samples at 595 nm.

The 5-larvae/pool samples were homogenized (using a Branson 250 Sonifier, Branson Ultrasonics Corporation, Danburry, Connecticut, USA) on ice on potassium phosphate buffer 0.1 M, pH 6.5, centrifuged (4 °C, 10,000× *g*, 5 min), and the supernatant used for enzymatic determination. GST was determined using 0.05 mL of homogenate and 0.250 mL of the reaction mixture (reduced 10 mM glutathione (GSH) and 60 mM 1-chloro-2.4-dinitrobenzene), and the activity was determined at 340 nm by measuring the increase in the absorbance based on the method described in Habig et al. (1974) [35]. ChE was determined at 414 nm using 0.05 mL of homogenate and 0.250 mL of the reaction solution (containing 10 mM 5:5-dithiobis-2-nitrobenzoic acid solution with sodium hydrogen carbonate, 0.075 M acetylcholine) according to Ellman et al. (1961) [36]. LDH determination used 0.04 mL of the sample, 0.250 mL of NADH (4 mM) and 0.04 mL of pyruvate (10 mM), and the activity was measured at 340 nm by continuously monitoring the oxidation of NADH, reflected in the decrease in absorbance [37]. CAT activity was measured using 0.02 mL of homogenate and monitoring the decrease in H_2_O_2_ at an absorbance of 240 nm [38].

The 15-larvae/pool samples were homogenized on ice on potassium phosphate buffer containing a 19 mM KH_2_PO_4_, 30 mM K_2_HPO_4_, 0.5 mM EDTA, and 1% protease inhibitor cocktail. The samples were homogenized and centrifuged at 15,000× *g* for 15 min at 4 °C. GR activity was measured at 340 nm after adding to 0.03 mL of the sample 0.2 mL of 0.3 mM NADPH and 0.01 mL of 22 mM GSSG [39]. GPx activity was measured at 340 nm after adding to 0.03 mL of the sample 0.02 mL of assay solution (5 mM NaN3, 18 mM GSH, 0.9 U/mL GR), 0.01 mL of 6 mM NADPH, and 0.01 mL of 0.016% H_2_O_2_ [39].

### 2.7. Data Analyses

Graphical plotting and statistics were performed using IBM SPSS Statistics (IBM^®^ SPSS^®^ Statistics 29.0). Normal data sets were analyzed by one-way ANOVA followed by the Dunnett’s multiple comparison test. For data sets not following a normal distribution, the nonparametric Kruskal–Wallis test was performed followed by the Dunn’s multiple comparison procedure. Logistic regression models were adjusted to mortality and hatching data to calculate L(E)C_50_ values using SigmaPlot v14. A significance level of 0.05 was considered.

## 3. Results

### 3.1. Cyanobacteria Extract—Site 1 and Site 2

#### 3.1.1. Mortality and Hatching

At 96 h of exposure, mortality presented a dose-response increase at both sites (Figure 2) reaching 60% at site 1 and 38% at site 2 at the highest concentration tested. For site 1, it was possible to calculate an LC_50_ value of 567.1 mg/L (standard error (ste) = 358).

Embryos’ hatching (Figure 2) was significantly affected during exposure to extracts from site 1. At 48 h, an induction at the lowest concentration (144.90 mg/L) and an inhibition at the higher concentrations tested (325.90, 488.90, 733.30, and 1100 mg/L) was observed, although differences among treatments were not detected by the multiple comparison test. At 72 h, a dose-dependent inhibition was observed with an LC_50_ value of 349.6 mg/L (ste = 23.78). At 96 h, the inhibition was still evident with only 20% of the embryos hatched at the two highest concentrations. Extracts from site 2 did not elicit significant delays on hatching and after 96 h of exposure, almost 100% of the embryos had hatched, even at the highest concentrations.

#### 3.1.2. Development Delay

Extracts from site 1 induced a significant developmental delay in zebrafish embryos compared to the control group (Figure 3). At 48 h of exposure, the embryos showed a delay in development at all concentrations except the lowest. This delay was dose-dependent with embryos exposed to the highest concentration (1100 mg/L) being arrested in the 10 hpf stage. At 72 h of exposure, the embryos exposed to the lowest concentration presented normal development while the most delayed embryos at the highest concentration were still at the 13 hpf stage. At 96 h of exposure, the developmental stage of the embryos at the highest concentration was comparable to the 28 or 36 hpf stages. Finally, at 120 h of exposure, a delay could still be seen at concentrations equal or above 325.9 mg/L. The maximum delay could be seen at the highest concentration where the majority of embryos presented a development equivalent to 48 hpf. The embryos exposed to the extract from site 2 showed a nonsignificant effect in their developments during the entire exposure.

#### 3.1.3. Locomotor Behavior

Locomotor behavior was not analyzed at site 1 for the concentration of 1100 mg/L due to significant mortality (>50%) recorded for this concentration. Effects in the swimming performance were more evident at site 1 where a dose-dependent decrease in the swimming distance (the highest concentration presented 35% of the control activity) and in the distance traveled in slow movements was observed (Figure 4). Extracts from site 1 did not elicit a clear effect in the path angles of embryos’ movement, although some significant effects were observed for sporadic concentrations/angles classes. Similarly to site 1, exposure to extracts from site 2 also caused a decreased movement measured as swimming distance (embryos exposed to 733.3 mg/L presented 85% of the control activity) and distance traveled in slow movements, although not as intense as for site 1 and not verified for all concentrations. Regarding path angles, exposure to extracts from site 2 caused a decrease in the proportion of Class 1 angles and an increase in the proportion of Class 4 angles (Figure 4).

#### 3.1.4. Biochemical Determinations

Exposure to the extracts did not significantly change the activity of the biotransformation enzyme, GST, for any of the sites (Figure 5). Regarding CAT, GPx, and GR, enzymes more directly involved in the antioxidant system, extracts from site 1 clearly inhibited CAT and an increase and decrease trend can be observed for GPx and GR, respectively. No effects were observed in the activities of the antioxidant enzymes of embryos exposed to cyanobacteria extracts from site 2. ChE, a marker of neurotoxicity, was inhibited at both sites at the highest concentrations. LDH was inhibited at site 1 (at the two highest concentrations) but not at site 2 (Figure 5).

### 3.2. Filtrates—Site 1 and Site 2

#### 3.2.1. Mortality and Hatching

Filtration of the cyanobacterial samples resulted in filtrates with a strong color as follows: a dark blue color at site 1 and a light/bright blue color at site 2. Exposure to the filtrates elicited effects at mortality and hatching levels of zebrafish embryos at both sites (Figure 6). At site 1, a dose-dependent increase in the mortality was observed with the highest concentration leading to 100% of mortality. In site 2, for the same range of concentrations, the maximum mortality recorded was ~30%. Regarding embryo hatching at 48 h, for both sites, lower concentrations led to higher hatching rates than the control while the highest concentrations inhibited the hatching. Although no significant effects could be seen when comparing the several treatments against the control, multiple comparisons showed the differences between lower and higher concentrations. At 72 h of exposure, the hatching rate was close to 100% in all treatments except for 2.86 and 10% in site 1. In these treatments, the hatching rate was approximately 50 and 0%, respectively.

#### 3.2.2. Locomotor Behavior

Behavioral analysis was not carried out at the two highest concentrations (2.86% and 10.00%) at site 1 due to a high mortality elicited by these concentrations (>50%). The analyzed parameters were swimming distance, the distance traveled in slow and rapid movements, and path angles (Figure 7). Filtrates obtained from site 1 elicited no behavioral changes on zebrafish larvae, although a slight decreasing trend can be observed in the distance traveled in slow movements. Filtrates obtained from site 2 seemed to decrease the swimming distance (not significant) and the distance traveled by larvae in slow movements (significant differences toward the control at 10%). No differences were detected in the proportions of angles classes after exposure to site 2 filtrate.

#### 3.2.3. Biochemical Determinations

Enzymes’ activities of zebrafish larvae exposed to filtrates from site 1 and 2 did not respond as clearly as larvae exposed to the extracts (Figure 8). At site 1, GST activity in larvae exposed to different concentrations of filtrates showed no significant differences compared to the control, while on site 2, the 0.80% concentration showed a statistically significant decrease compared to the control. The activities of the enzymes from the antioxidant system (CAT, GPx, and GR) were not modified in any of the sites when compared to the control activities. ChE activity shows a decreasing trend at both sites, with significant differences compared to the control in 0.8% and 2% concentrations at site 1. At site 2, however, ChE activity inhibition was not statistically significant. LDH activity seemed to be unaffected in the larvae exposed to different concentrations of filtrates in any of the sites when compared to the control.

### 3.3. Water Column—Site 1 and Site 2

In the assay performed with the samples from the water column of both sites, there were no differences in terms of mortality or embryonic development compared to the control. Therefore, behavioral and biochemical analyses were not carried out because no effects were expected.

## 4. Discussion

### 4.1. Toxicity of Cyanobacteria Extract

The early life stages of fish and other aquatic organisms can be particularly susceptible to environmental stressors, such as toxic cyanobacterial blooms [41,42,43,44]. The cyanobacteria present in the bloom studied in this work were identified as *Microcystis aeruginosa*, one of the most common species and responsible for the production of microcystins (MCs), toxins known to be toxic to aquatic organisms such as fish, causing a decrease in survival, disruption of development, neurotransmitters alteration, and induction of oxidative stress as demonstrated, for example, by [42,43,45]. In the present study, the cyanobacteria extracts tested elicited important effects at the different levels assessed. Literature suggests that pure cyanotoxins do not have a high acute toxicity for zebrafish embryo/larvae in similar exposure scenarios. For instance, Refs. [42,46] showed that no mortality occurred in zebrafish larvae exposed for 4 days to 500 μg/L or for 7 days to 600 μg/L of MCLR respectively, while [47], despite having tested a higher concentration of MCLR (5000 μg/L) for 4 days, showed that the survival rate was only reduced by 7%. In our work, although effects on survival were observed, direct comparison with studies using pure cyanotoxin is not possible, as no direct quantification of the cyanotoxin was made. However, some authors, such as [48,49], claimed that crude extracts from the batch cultures and field samples present a higher toxicity when compared to embryos exposed to pure MCLR. This is because cyanobacterial extracts from blooms may contain a greater amount and variety of cyanotoxins; moreover, the presence of other substances such as nutrients, inorganic elements, and even viruses and bacteria in the crude extracts may also contribute to increase the absorption and toxicity of the toxins, by synergetic mechanisms [46,50,51]. In the present study, samples preparation followed the methodology used by [30], where crude extracts from *Raphidiopsis raciborskii* strains were tested eliciting embryo mortality in the first 24 h along with a developmental delay that followed a dose response curve, which was also observed in the present study.

Hatching effects observed at 48 hpf revealed a biphasic pattern, showing an increase in the hatching rate at the lowest concentration of cyanobacteria extracts and a decrease at the highest. Early hatching was not an expected mechanism since, in general, exposure to cyanotoxins causes a delay in the development of aquatic organisms [43,48,52,53]. However, this premature hatching has already been recorded in other studies in response to sublethal doses of cyanotoxins. In [49], earlier hatching occurred in rainbow trout exposed to MC-RR, MC-YR, and MC-LR. Jacquet et al. [52] also recorded a premature hatching of 2 to 3 days in medaka fish embryos microinjected with MCLR and [46] recorded an early hatching of zebrafish embryos exposed to MCLR. In our study, early hatching was not related to an eventual advance on embryo development because early hatched embryos showed normal development comparable to the control. Alternatively, low concentrations of cyanotoxins may lead to the softening and rupture of the chorion, allowing the embryos to hatch earlier. This may be due to the induction of the enzyme chorionase, responsible for the rupture of the chorion. This induction at low concentrations may be a form of phenotypic plasticity in response to stress, where organisms accelerate their life cycle as a survival strategy [54].

The embryos exposed to the extract from site 1 suffered a marked developmental delay with hatching inhibition at all concentrations, except for the concentration that promoted hatching (144.90 mg/L). This was verified in other studies where zebrafish embryos were exposed to crude extracts [30,48,49]. Development delay may be the consequence of cellular malfunction and cellular death caused by the presence of ROS and oxidative stress. As it will be further discussed, this is partly supported by our biochemical data that suggest a response of the antioxidant system. On the other hand, cyanotoxins, especially MCs, are known to inhibit protein phosphatase 1 (PP1) and protein phosphatase 2 (PP2), which play a fundamental role in embryonic development [55].

When an organism is exposed to an environmental stress, its behavior may be changed as a defense response [56,57]. The impact of MCs exposure on fish behavior is not yet properly characterized but evidence shows that MCs can accumulate in brain cells, affecting the nervous system and leading to behavioral changes [53,58,59]. A sudden transition to darkness elicits a startle response in zebrafish larvae, characterized by a transient burst of activity [60]. The analysis of this response is a noninvasive method that can be used to assess visual function, development of the nervous system, and the development of the locomotor system [61]. The results for site 1 show a general trend for reduction in the locomotion of larvae or hypoactivity after exposure to the extracts, with larvae from the two highest concentrations swimming half of the distance than the control larvae. The same type of effects was reported by [42] that showed a weakened response to sudden darkness and hypoactivity in zebrafish larvae exposed to MC-LR; these effects were attributed to functional motor neuron and/or skeletal muscle defects. This suggest a neurotoxic action of MCs that can be explain by disruption of the cholinergic system as supported by the ChE activity decrease observed in the present work. Qian et al. [62] showed a decreased locomotor activity in newly hatched zebrafish exposed to *M. aeruginosa*, which was correlated with a decrease in the dopamine levels. In addition, exposure to developmental neurotoxicants during embryogenesis may also induce hypoactivity in zebrafish larvae, since when acting during critical stages of embryonic development, these substances may lead to a disruption of the normal development and function of the nervous system [63,64]. It is also important to highlight that at the highest concentration studied for behavioral effects (733.3 mg/L), the embryos were considerably delayed in their development, and then, it is likely that behavioral effects detected were due not only to direct neurotoxicity but also to differences in locomotor performance caused by development delay. On the other hand, Ref. [65] recorded hyperactivity in zebrafish embryos after acute exposure to cyanobacterial extracts, suggesting an effect dependent on the fish strain, exposure scenarios, and/or on the tested concentrations. The measurement of the path angles is useful to show changes in swimming patterns of the organisms; however, for site 1, for larvae exposed to the extracts, no clear effects were observed.

Cellular biomarkers are sensitive tools that can be used as early indicators of toxicity. In this study, an increase in the activity of the enzymes of the antioxidant system was expected, since the induction of oxidative stress by cyanotoxins is well documented [66,67,68]. However, the most evident response among the antioxidant enzymes analyzed was a decrease in CAT activity (especially at site 1). CAT is an important antioxidant enzyme present in most living tissues, acting as an antioxidant in the first line of defense of organisms [69]. Although an activity increase was expected, the response of antioxidant enzymes depends on the intensity of oxidative stress and, therefore, under extreme conditions, enzymes can exceed their catalytic/transforming capacity, leading to proteins, lipids, and DNA damage [70]. Thus, the decrease in CAT activity in this study may indicate oxidative damage. Although the results for GPx and GR were not very clear, in general, the results suggest that the oxidative system has been affected. Finally, it would have been important to measure the activity of LPO (lipid peroxidation) to understand the extent of the oxidative stress inflicted on the larvae.

Neurotransmitters such as acetylcholine are crucial for the transmission of signals across synapses. Disruption of neurotransmission pathways can lead to the impairment of specific nervous system functions [53,59]. Exposure to cyanobacterial extracts led to a reduction in ChE activity in zebrafish larvae at both sites, clearly indicating an effect on the cholinergic system. Usually, cyanotoxins are not known as cholinesterase inhibitors. For instance, Ref. [53] found that exposure to MC-LR resulted in the up-regulation of ChE gene expression and increased ChE activity, leading to a reduction in Ach content in larval zebrafish, contributing to altered behavioral responses. However, other reports on literature account for decreases in ChE activity such as in [62] after exposure of zebrafish embryos to *M. aeruginosa* and in [71] after sublethal exposure of *Pseudosida ramosa* to *Anabaena spiroides* extract (anatoxin-a). The ChE activity reduction observed in this study confirms the anticholinergic action of cyanotoxins of the *Microcystis* genus and is likely to be correlated with the locomotor reduction observed in the behavioral analysis.

An increase in LDH activity would be expected due to greater energy production in response to exposure to a stressor [72]. However, LDH activity showed a tendency to decrease, being significantly inhibited at the highest concentrations of the extract from site 1, contrary to what was observed in [73], in which an increase in LDH activity was observed in response to intraperitoneal administration of pure microcystin LR in silver carp. Since it plays a key role in cell metabolism, inhibiting LDH activity can have several implications, such as during embryonic and larval development, fish depend on an efficient metabolism for proper growth and development, which can be affected by low LDH levels. Moreover, low LDH levels can indicate a lower ability of the fish to metabolize and eliminate environmental toxins, leading to an impaired response to stress factors.

Comparing the responses obtained for the two sampling sites, we can conclude that although the same type of effects was observed, their intensity differed significantly among sites. The samples from site 1 showed a higher toxicity than the samples from site 2, especially attending to mortality, hatching, and development endpoints. This suggests that the visual differences recorded among sites (site 1—higher density and pastier in appearance, with brownish green coloration, looking like an older formation, site 2—more diluted in appearance, with bright green coloration, looking like a younger bloom) corresponded to different stages of the bloom life cycle also characterized by a different amount of cyanotoxins produced and different toxicities. Indeed, cyanobacteria in the bloom, after a phase of exponential growth, senesce and die or enter a state of metabolic dormancy [74]. The collapse of the bloom may be due to changes in the environment caused by the excessive growth of the bloom itself (such as nutrient depletion (mainly phosphorus and nitrogen), increased temperature, decreased oxygen, changes in light intensity, and water flow). In this case, these changes may constitute a stress factor to cyanobacteria, triggering the production of toxins as a defense mechanism and, thus, explaining the higher toxicity of older cells as verified in this work.

The collapse of blooms may also be due to predatory bacteria capable of secreting lysing agents [75], and viral lysis [76]. Considering that most cyanotoxins remain intracellular, a collapse of a bloom through bacterial or viral lysis will release a large amount of cyanotoxins into the water column and, thus, it is not surprising that water surrounding the cyanobacteria as represented by “the filtrates” in this work also present a higher toxicity in older areas of the bloom.

### 4.2. Filtrates

The aim of testing this fraction was to assess the toxicity associated with the water surrounding the cyanobacteria and that also contains cyanotoxins released from the cyanobacteria cells as opposite to the cyanotoxins enclosure in cyanobacteria cells themselves. Both filtrates revealed to be extremely toxic, particularly at site 1, where a concentration of only 10% of the original filtrate led to a 100% mortality of zebrafish embryos after 96 h of exposure. We cannot exclude that this high toxicity is partly explained by the fact that the water samples from the bloom were frozen before processing. In fact, the freezing and thaw process provokes cell lysis eliciting the release of the cells’ contents adding to the baseline toxicity of this fraction. Moreover, this fraction may also contain a multitude of other chemical stressors such as pollutants presenting toxicity or interacting synergistically with the cyanotoxins.

Regarding the hatching rate of embryos, interestingly, the biphasic response, or bell-shaped pattern that could be somewhat perceived in the extracts was, in the filtrates, very clear for both sites, with low concentrations inducing hatching and high concentrations delaying it. In addition to hatching, surprisingly, the other sublethal parameters assessed were not so responsive. In the case of locomotor behavior, only site 2 showed a decrease in the distance traveled in slow movements, confirming the trend previously observed for the extracts. Among biochemical responses, only ChE activity was decreased at site 1 (although the same trend can also be observed at site 2), also confirming the results obtained for the extracts.

In conclusion, following the pattern of the extracts, filtrates from site 1 presented more toxicity than site 2, confirming the hypothesis that the two sites represent two stages of the bloom life cycle.

## 5. Conclusions

After observing a bloom of cyanobacteria with peculiar characteristics in a city park in Aveiro (Portugal), it was decided to take samples from two sites with different stages of population evolution and test them. The samples were revealed to contain cyanotoxins eliciting developmental (early hatching for low concentrations, delayed hatching for high concentrations, and delayed development), behavioral (hypoactivity), and biochemical (ChE activity reduction and interference with the oxidative stress system) effects in zebrafish embryo/larval stages. This work emphasizes that the occurrence of cyanobacterial blooms in urban environments that can be highly toxic, including on a neurological level, is an effect that is not usually associated with environmental samples with a high content of MCs. In a climate change context where a higher frequency of cyanobacterial proliferation is expected, this topic should be properly addressed by competent entities to avoid deleterious effects on the biodiversity of urban parks and poisoning events of pets and people. Moreover, the two sites sampled were associated with different degrees of toxicity, suggesting that during the development of a bloom, a gradient of toxicity is being created in the water body.

## Figures and Tables

**Figure 1 microorganisms-12-02003-f001:**
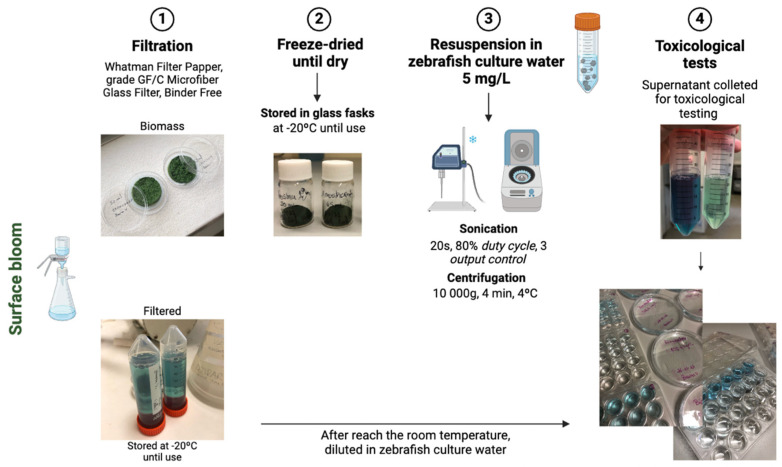
Schematic representation of the steps involved in the preparation of cyanobacterial extracts and filtered samples. Methodology based on [30].

**Figure 2 microorganisms-12-02003-f002:**
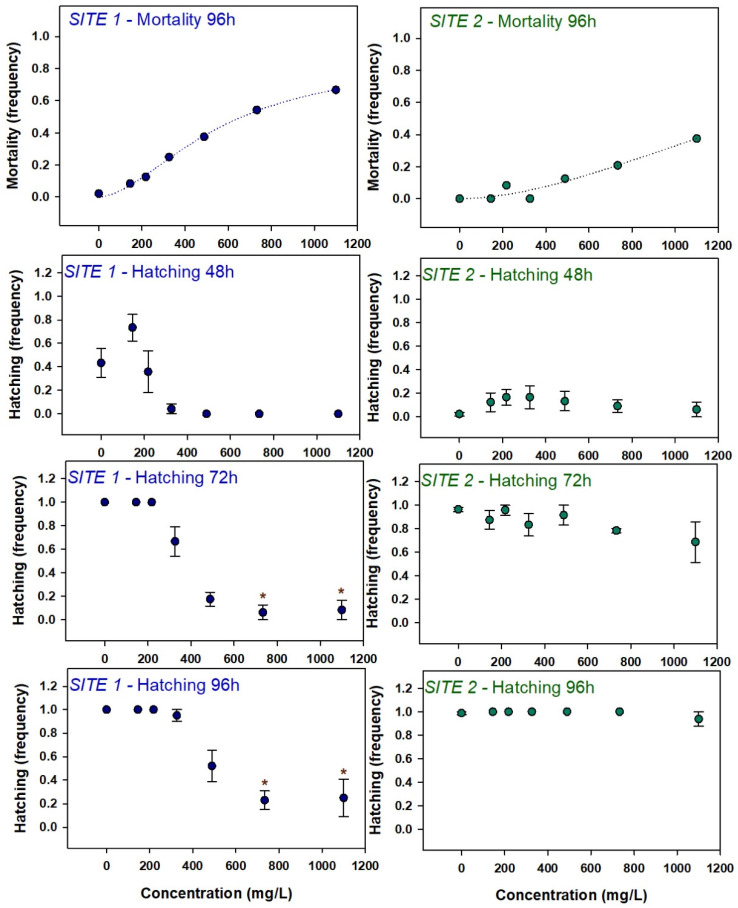
Mortality and hatching effects in zebrafish larvae exposed to cyanobacteria extracts. Mortality data was fitted to a logistic curve and symbols represent mean values. Hatching data are represented as mean values ± standard error. Asterisks denote significant differences toward the control indicated by the Dunn’s method.

**Figure 3 microorganisms-12-02003-f003:**
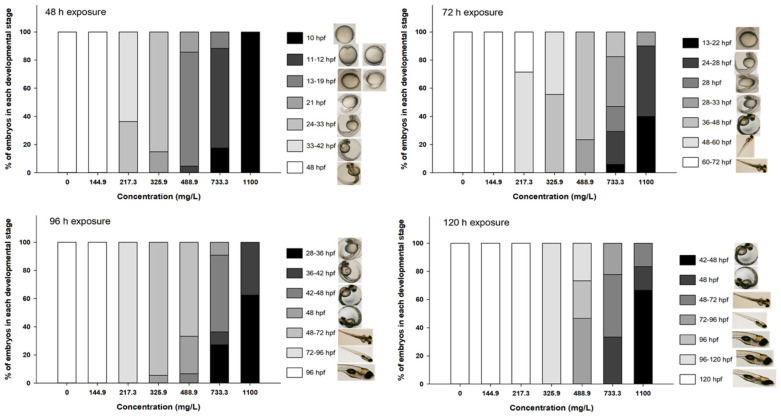
Developmental effects in zebrafish embryos/larvae exposed to cyanobacteria extracts from site 1. The legend indicates the stage(s) of development in hours (hours post-fertilization (hpf)) according to Kimmel et al. (1995) [31]. The images in the legend depict the respective stage or a representative stage within the indicated interval and were adapted from [40].

**Figure 4 microorganisms-12-02003-f004:**
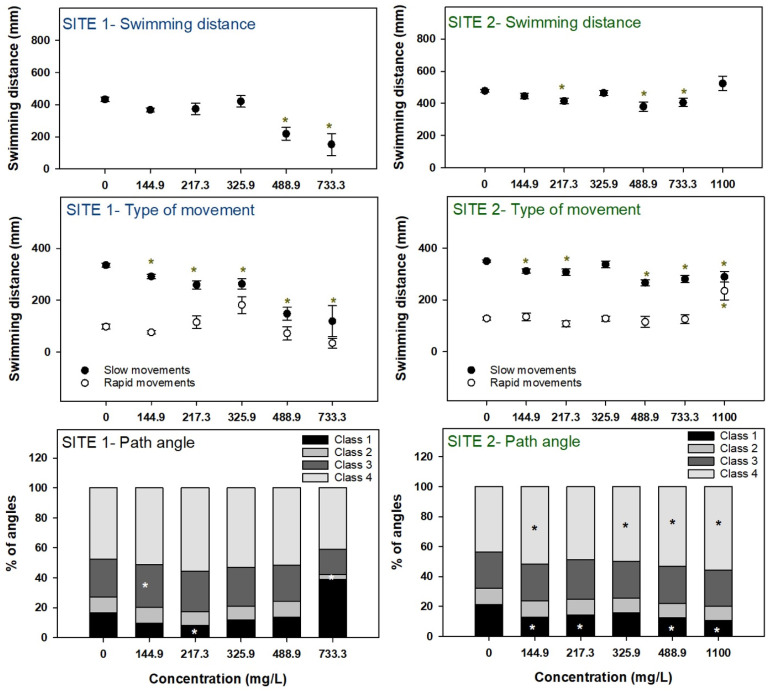
Behavioral effects in zebrafish embryos/larvae exposed to cyanobacteria extracts. Symbols represent mean values ± standard error. Bars represent mean values. Asterisks denote significant differences toward the control indicated by the Dunn’s method.

**Figure 5 microorganisms-12-02003-f005:**
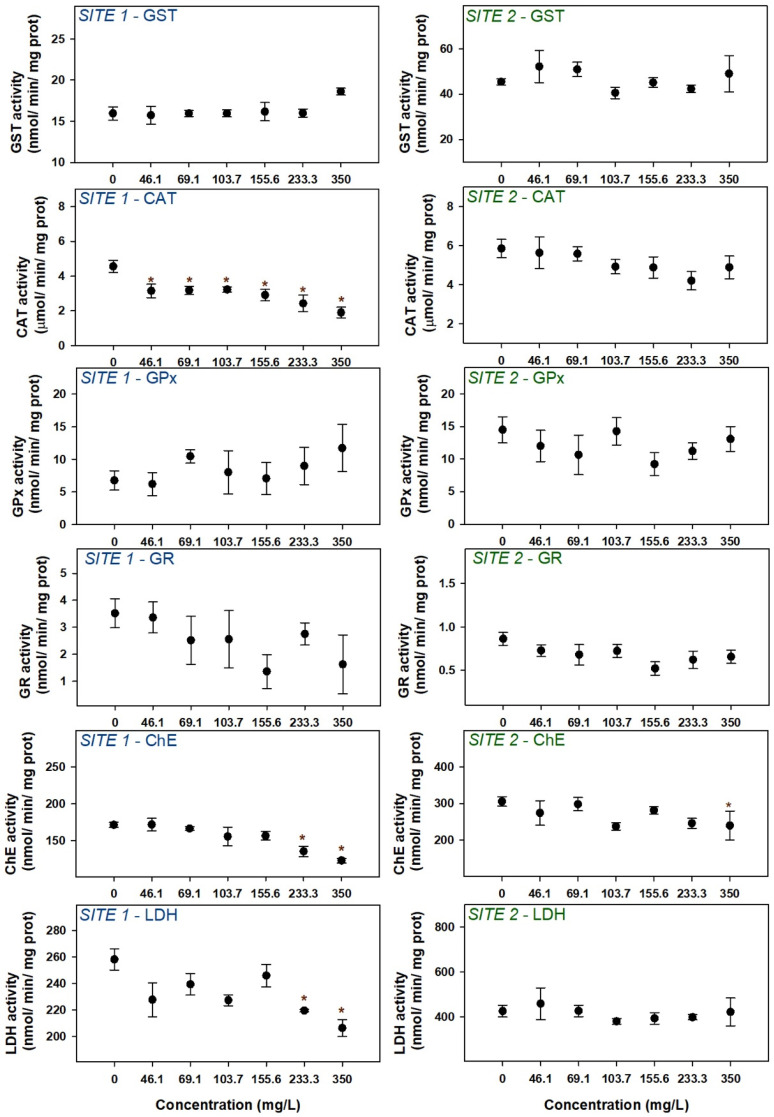
Biochemical effects in zebrafish embryos/larvae exposed to cyanobacteria extracts. Symbols represent mean values ± standard error. Asterisks denote significant differences toward the control indicated by the Dunnett’s method (site 1—ChE and LDH) or Dunn’s method (site 2—ChE).

**Figure 6 microorganisms-12-02003-f006:**
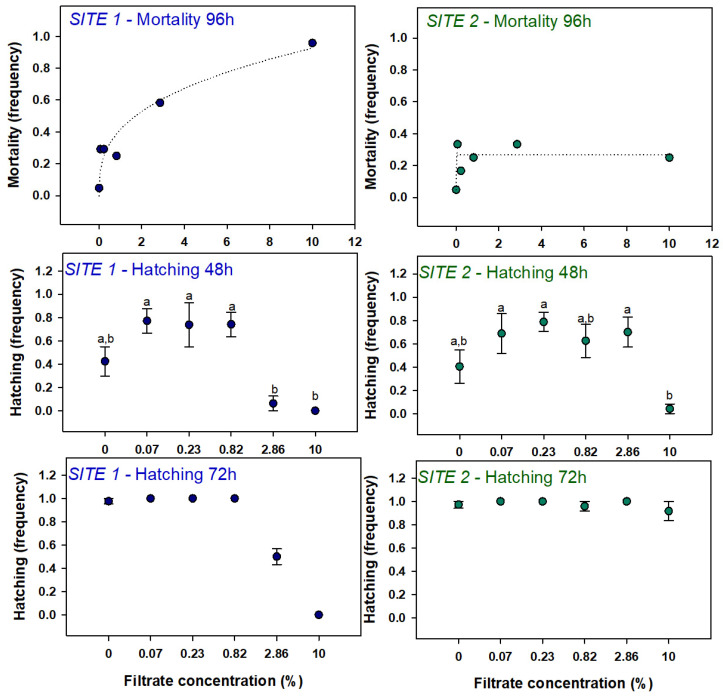
Mortality and hatching effects in zebrafish larvae exposed to filtrate. Mortality data was fitted to a logistic curve and symbols represent mean values. Hatching data are represented as mean values ± standard error. Different letters above the symbols denote significant differences between treatments indicated by the Dunn’s method.

**Figure 7 microorganisms-12-02003-f007:**
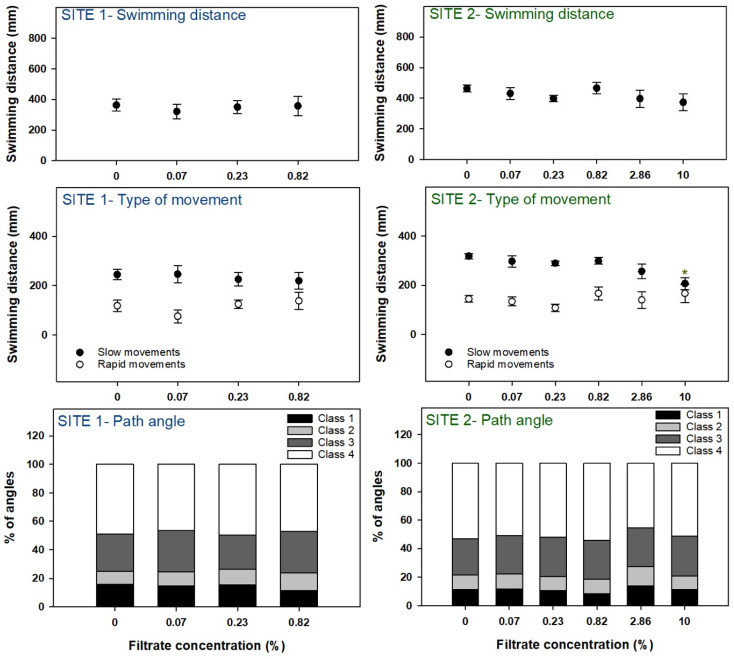
Behavioral effects in zebrafish embryos/larvae exposed to cyanobacteria filtrates. Symbols represent mean values ± standard error. Bars represent mean values. Asterisks denote significant differences toward the control indicated by the Dunn’s method.

**Figure 8 microorganisms-12-02003-f008:**
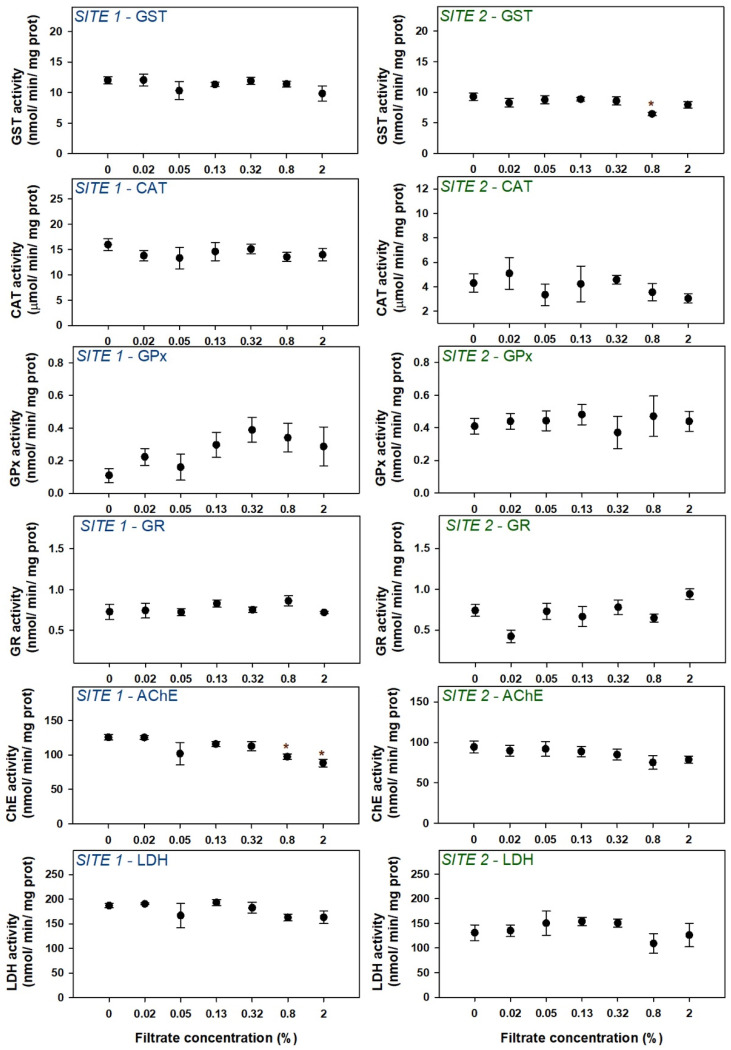
Biochemical effects in zebrafish embryos/larvae exposed to cyanobacteria extracts. Symbols represent mean values ± standard error. Asterisks denote significant differences toward the control indicated by the Dunn’s method.

## Data Availability

The original contributions presented in the study are included in the article/Appendix A, further inquiries can be directed to the corresponding authors.

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
