# Peer review of "Cyanobacterial Blooms in City Parks: A Case Study Using Zebrafish Embryos for Toxicity Characterization"

_microorganisms, 2024, doi:10.3390/microorganisms12102003_

Round 1
Reviewer 1 Report
Comments and Suggestions for Authors
The manuscript entitled “Cyanobacterial blooms in city parks: a case study using zebrafish embryos for toxicity characterization » is devoted to Harmful Algae Blooms (HAB) which is an important problem due to its negative socioeconomic impacts. Authors describe the toxicity of a cyanobacterial bloom from a urban park in the city of Aveiro (Portugal) using different standard techniques. To my mind this manuscript is topical and corresponding to the aims and scopes of the “Microorganisms” journal. I am ready to recommend it for publication after correcting several comments.
1. The authors should correct the abstract of the manuscript. It should contain brief and specific results of the study, and not a description of the problem the authors worked on.
2. The introduction also seems to me to contain a lot of unnecessary general information, for example lines 26-32, 74-76 79-92
3. The purpose of the study should be formulated at the end of the introduction.
4. Since only 2 samples were collected, their relevance for the study and formulation of the conclusions obtained should be formulated.
5. The phrase The taxonomic study was carried out using light microscopy (Olympus CX31) and the flora is not clear. The taxonomic studies should be separately identified and described in more detail. Based on the results, the presence of only one organism in bloom seems somewhat dubious.
6. It is not described how the samples were collected from the water column below and how they were used in the study. Were there cells in these samples and how many.
7. Is optical microscopy sufficient to identify an organism to the species level? It seems to me that more evidence should have been provided. This is important for the journal Microorganisms.
8. 499-501 should be moved to the Materials and Methods section
9. It is necessary to clarify the phrase site confirming the hypothesis that the two sites represent two stages 520 of the bloom life cycle. Why? Is it related to depth? Were the samples collected at the same temperature and at the same time?
10. globally, since the authors only collected 2 samples, it is necessary to very clearly explain the difference between them and the differences between them should be in red line throughout the text.
11. As a recommendation, I would still like to see an analysis of toxins with their identification and concentration characteristics.
12. in conclusion it is worth writing not about what was done in the work, but what the results led to. What new things they gave, what further prospects for fundamental and applied science
Comments on the Quality of English LanguageMinor editing of English language required.
Author Response
Thank you for your comments and your help in improving the quality of the manuscript.
Comments 1: The authors should correct the abstract of the manuscript. It should contain brief and specific results of the study, and not a description of the problem the authors worked on.
Response 1: We agree with this comment. Therefore, we add values to the abstract. Page 1, line 12-21:
With the increasing global temperatures, urban parks, renowned for their multifaceted contributions to society, have been largely affected by blooms of toxic cyanobacteria. In this work, the toxicity of two different stages of development of a cyanobacterial bloom from a city park was assessed, evaluating mortality, hatching, development, locomotion (total distance, slow and rapid movements, and path angles) and biochemical parameters (oxidative stress, neurological damage, and tissue damage indicators) in zebrafish embryos/larvae (Danio rerio). Results showed significant effects for the samples with more time of evolution at developmental level (early hatching for low concentrations (144.90 mg/L), delayed hatching for high concentrations (significant values above 325.90 mg/L) and delayed development at all concentrations), behavioural level (hypoactivity) and biochemical level (Cholinesterase (ChE) activity reduction and interference with the oxidative stress system for both stages of evolution).
Comments 2: The introduction also seems to me to contain a lot of unnecessary general information, for example lines.
Response 2: We agree to this and made changes, removing unnecessary information, such as what was written in lines 28-34 and 76-106, pages 1 and 2:
Comments 3: The purpose of the study should be formulated at the end of the introduction.
Response 3: Thank you for pointing this. We agree and reformulated the objective at the end of the introduction, as can be seen on page 3, lines 116-123:
The main objective of this work was to evaluate the toxicity of a cyanobacterial bloom in a lake of an urban park in the city of Aveiro (Portugal) and to understand whether different stages of evolution of a bloom represent different toxicities. Using embryos of zebrafish, a multilevel assessment was performed to evaluate mortality, hatching, development, biochemical (Glutathione S-transferase (GST), Catalase (CAT), Glutathione Peroxidase (GPx), Glutathione Reductase (GR), Cholinesterase (ChE) and Lactate dehydrogenase (LDH)) and behavioral (total distance, slow and rapid movements, and path angles) parameters.
Comments 4: Since only 2 samples were collected, their relevance for the study and formulation of the conclusions obtained should be formulated.
Response 4: We have reformulated the conclusions section, where we show the relevance of the two samples and the conclusions for this study, page 17, lines 566-575:
After observing a bloom of cyanobacteria with peculiar characteristics in a city park in Aveiro (Portugal), it was decided to take samples from two sites with different stages of population evolution and test them. The samples revealed to contain cyanotoxins eliciting developmental (early hatching for low concentrations, delayed hatching for high concentrations and delayed development), behavioural (hypoactivity) and biochemical (ChE activity reduction and interference with the oxidative stress system) effects in zebrafish embryo/larval stages. This work emphasizes that the occurrence of cyanobacterial blooms in urban environments can be highly toxic, including on a neurological level, an effect that is not usually associated with environmental samples with high content of MCs.
Comments 5: The phrase The taxonomic study was carried out using light microscopy (Olympus CX31) and the flora is not clear. The taxonomic studies should be separately identified and described in more detail. Based on the results, the presence of only one organism in bloom seems somewhat dubious.
Response 5: Thank you for pointing this. We agree that the phrase was not clear and reformulated it, page 3, lines 134-137:
The bloom samples were observed under a light microscopy (Olympus CX31), and the taxonomic study was based on Komárek and co-autors work ([31]), identifying the presence of Microcystis aeruginosa.
While monitoring the evolution of the sample, we noticed the disappearance of other organisms that were not present during the toxicity tests. No other species, particularly of cyanobacteria, was identified. Some colonies of the samples of site 2 had diatoms in the colony mucilage.
Comments 6: It is not described how the samples were collected from the water column below and how they were used in the study. Were there cells in these samples and how many.
Response 6: We describe how the sample was collected in the materials and methods, section collection of samples: page 3, lines 126-133: The cyanobacterial material from the surface of the bloom (3 cm depth), as well as the respective water below the cyanobacteria, to a depth of 50 cm.
We added a sentence saying that no cells were observed in the water column sample, page 3, lines 137-138: All the samples were collected at the same time and temperature, directly into PET bottles. The bloom samples were observed under a light microscopy (Olympus CX31), and the taxonomic study was based on Komárek and co-autors work ([31]), identifying the presence of Microcystis aeruginosa. No cells were observed in the water column samples. Immediately after the identification, the samples were frozen at -20ºC.
Comments 7: Is optical microscopy sufficient to identify an organism to the species level? It seems to me that more evidence should have been provided. This is important for the journal Microorganisms.
Response 7: In addition to the aspects observable under light microscopy, and specifically the morphology of the colony, the cells and their cytology, the ecological component must also be considered. With a view to confirmation, molecular biology methodologies (eg. genetic methodologies) should be used.
Comments 8: 499-501 should be moved to the Materials and Methods section.
Response 8: We did not agree that lines 499-501 should be moved, as they are essential to the discussion.
Comments 9: It is necessary to clarify the phrase site confirming the hypothesis that the two sites represent two stages 520 of the bloom life cycle. Why? Is it related to depth? Were the samples collected at the same temperature and at the same time?
Response 9: The samples were collected at the same depth, at the same time, on the same day and at the same temperature, which has been added to the text, page 3, line 133: All the samples were collected at the same time and temperature, directly into PET bottles.
Comments 10: globally, since the authors only collected 2 samples, it is necessary to very clearly explain the difference between them and the differences between them should be in red line throughout the text.
Response 10: One of the samples (site 1) corresponds to a more advanced state of development, which is not observed in an initial state (site 2) of the population dynamics of the reported species.
Comments 11: As a recommendation, I would still like to see an analysis of toxins with their identification and concentration characteristics.
Response 11: Thank you for pointing this. We apologise that the quantification of the samples was not included in the manuscript by mistake. We have added a section on materials and methods, quantification, page 4, lines 166-176:
2.3. Quantification
To determine the group and concentration of cianotoxins in the bloom and water column samples, the Microcystin-ADDA ELISA Kit (Enzo Life Sciences, ALX-850-319-KI01), an immunoassay for the quantitative detection of microcystins and nodularins in water samples through indirect competitive ELISA, was used according to the manufacturer’s instructions. For each bloom (site 1 and site 2) three dilutions were made: 5x, 10x and 50x. The water column samples from both sites were analyzed with 5x and 10x dillutions. The immunoassay showed us that the samples were composed by microcystins, no specific variant identified. However, the dilutions made were not sufficient for the environmental samples to fit the standards curve, which means that the concentration of microcystins of the samples were higher than the maximum concentration of the KIT standards (5.0 ppb).
Comments 12: in conclusion it is worth writing not about what was done in the work, but what the results led to. What new things they gave, what further prospects for fundamental and applied science.
Response 12: The relevance to the study centers on the evaluation of a specific sample with peculiar characteristics. We have rewritten a few sentences in the conclusion, emphasizing the fact that MCs have toxic effects on the neurological level, which is not normally associated with the action of MCs, page 17, lines 566-575:
After observing a bloom of cyanobacteria with peculiar characteristics in a city park in Aveiro (Portugal), it was decided to take samples from two sites with different stages of population evolution and test them. The samples revealed to contain cyanotoxins eliciting developmental (early hatching for low concentrations, delayed hatching for high concentrations and delayed development), behavioural (hypoactivity) and biochemical (ChE activity reduction and interference with the oxidative stress system) effects in zebrafish embryo/larval stages. This work emphasizes that the occurrence of cyanobacterial blooms in urban environments can be highly toxic, including on a neurological level, an effect that is not usually associated with environmental samples with high content of MCs.
Reviewer 2 Report
Comments and Suggestions for Authors
The study by Veira et al., titled “Cyanobacterial Blooms in City Parks: A Case Study Using Zebrafish Embryos for Toxicity Characterization,” investigated the toxicity of a cyanobacterial bloom of Microcystis aeruginosa from a city park in Portugal, evaluating mortality, hatching, development, locomotion, and biochemical parameters in zebrafish embryos/larvae (Danio rerio). Overall, the study presents appropriate and well-explained methodologies, coherent results with the approaches used, and a well-discussed comparison with similar investigations. For these reasons, I commend the authors for their effort and dedication in developing the study. However, despite the study’s merits, in my view, it needs to better highlight its importance and novelty, as the investigation of this topic has been extensively studied. Even though this is a case study, these aspects should be made clear to the reader. Additionally, the absence of cyanotoxin quantification represents a significant limitation of the investigation. Other minor issues need to be addressed to improve the quality of the manuscript, so I recommend major revisions.
Below are some individual points:
Introduction:
- It is important to better emphasize the novelty and significance of your study in the introduction. As currently written, the study seems like "just another" among the many that address this topic. Even though it is a case study, you can still highlight the unique qualities and approach of your investigation compared to other studies. This should be explored and made evident.
- The paragraphs are too long, making the text dense. This recommendation also applies to the discussion section. Shorter paragraphs will improve the flow and make the text less tiresome for readers.
- In the second paragraph, when discussing waste discharges, it would be relevant to mention that these situations intensify during drought periods when river flow decreases but the input of organic and inorganic waste from domestic and industrial sewage remains constant. You could also highlight that periodic droughts have intensified in many regions due to climate change.
- Lines 54-57: “Cyanotoxins may compromise the survival and fitness of various aquatic organisms, wild animals, as well as domestic animals and humans [23], [24]. The toxic effects of MCs can manifest in liver failure in aquatic organisms, wildlife, livestock, and humans.” This information seems repetitive. It should be rephrased for clarity and conciseness.
- Lines 71-102: This entire paragraph needs to be restructured. It is too long and tedious in its current form. It is important to present zebrafish as a well-established model, but avoid excessive discussion about "where it has been used previously." Be more concise in your statements. Additionally, briefly explain the experimental model and conclude by stating the objective of the study, which involves evaluating mortality, hatching, development, locomotion, and biochemical parameters.
Results:
- The section “3.1 Identification of Samples” would be better placed in the methodology, extending the text in section 2.1. You could mention that after sample collection, the samples were identified [...].
- Lines 288-291: “The following biomarkers were measured in the larvae exposed to different concentrations of cyanobacterial extract: Glutathione S-transferase (GST), Catalase (CAT), Glutathione Peroxidase (GPx), Glutathione Reductase (GR), Cholinesterase (ChE) and Lactate dehydrogenase (LDH).” This information has already been clearly stated in the methodology, so there is no need to repeat it here.
Discussion:
- I found the lack of a paragraph discussing the limitations of the study to be a significant omission. Particularly, the absence of quantitative measurements of cyanotoxins in the samples should be addressed, along with any other notable limitations.
- Lines 519-521: “In conclusion, following the pattern of the extracts, filtrates from site 1 presented more toxicity than site 2, confirming the hypothesis that the two sites represent two stages of the bloom life cycle.” Was this the hypothesis of the study? If so, this was not clearly presented earlier in the manuscript. If this was indeed the hypothesis, I recommend making it explicit in the introduction and methodology sections as well.
Other:
- Although the journal does not impose strict limitations on the number of references, 107 references for a case study seems excessive. While I appreciate that I did not identify any self-citations (which is positive), there are instances where the authors cite numerous studies to clarify a point that could be sufficiently addressed with 2 or 3 references. I suggest revisiting the references and aiming to reduce the overall number where possible.
Finally, I once again commend the authors for their study and the effort dedicated to its development. I hope that my suggestions, along with those from other reviewers, will help ensure that the manuscript reflects the quality of the effort invested.
Best regards,
Author Response
Thank you for your comments and your help in improving the quality of the manuscript.
Comments 1: It is important to better emphasize the novelty and significance of your study in the introduction. As currently written, the study seems like "just another" among the many that address this topic. Even though it is a case study, you can still highlight the unique qualities and approach of your investigation compared to other studies. This should be explored and made evident.
Response 1: In the introduction, we discussed the importance of urban parks, page 2, lines 71-75.
In this case study, we wanted to test samples from a bloom of cyanobacteria in different states of evolution rather than isolated cyanotoxins, as most case studies report. We reformulated the final paragraph of introduction, to be more clear, page 3, lines 116-123:
The main objective of this work was to evaluate the toxicity of a cyanobacterial bloom in a lake of an urban park in the city of Aveiro (Portugal) and to understand whether different stages of evolution of a bloom represent different toxicities. Using embryos of zebrafish, a multilevel assessment was performed to evaluate mortality, hatching, development, biochemical (Glutathione S-transferase (GST), Catalase (CAT), Glutathione Peroxidase (GPx), Glutathione Reductase (GR), Cholinesterase (ChE) and Lactate dehydrogenase (LDH)) and behavioral (total distance, slow and rapid movements, and path angles) parameters.
Comments 2: The paragraphs are too long, making the text dense. This recommendation also applies to the discussion section. Shorter paragraphs will improve the flow and make the text less tiresome for readers.
Response 2: Thank you for pointing this. We agree, specially for introduction section, and made some changes.
Comments 3: In the second paragraph, when discussing waste discharges, it would be relevant to mention that these situations intensify during drought periods when river flow decreases but the input of organic and inorganic waste from domestic and industrial sewage remains constant. You could also highlight that periodic droughts have intensified in many regions due to climate change.
Response 3: We agree that this can add value to what is being discussed, so we complement the sentence, page 2, lines 49-52:
Phosphorus and nitrogen concentrations have a strong influence on cyanobacterial growth, potentiated through industrial, domestic, and agricultural waste discharges [12], [13], which are intensified during periods of drought, when the river’s flow decreases but the input of organic and inorganic waste remains constant which, in a context of climate change, is an issue that deserves even more attention, since periods of drought have intensified.
Comments 4: Lines 54-57: “Cyanotoxins may compromise the survival and fitness of various aquatic organisms, wild animals, as well as domestic animals and humans [23], [24]. The toxic effects of MCs can manifest in liver failure in aquatic organisms, wildlife, livestock, and humans.” This information seems repetitive. It should be rephrased for clarity and conciseness.
Response 4: Thank you for pointing this. We totally agree, so we rephrase the sentence, page 2, lines 60-63:
Cyanotoxins may compromise the survival and fitness of various aquatic organisms, wild animals, as well as domestic animals and humans [16], [17], leading to symptoms such as liver failure [14], [18].
Comments 5: Lines 71-102: This entire paragraph needs to be restructured. It is too long and tedious in its current form. It is important to present zebrafish as a well-established model, but avoid excessive discussion about "where it has been used previously." Be more concise in your statements. Additionally, briefly explain the experimental model and conclude by stating the objective of the study, which involves evaluating mortality, hatching, development, locomotion, and biochemical parameters.
Response 5: We agree, so we reformulated the entire paragraph, page 3, lines 107-123:
In this study, the zebrafish (Danio rerio) was chosen as a model because of its value in toxicology, due to its rapid and cost-effective assessments of chemical hazards and environmental toxicity, as well as the possibility of assessing embryonic development, such as delays and malformations, due to embryo’s optical transparency. Moreover, the zebrafish embryo is considered an alternative model to animal experimentation and their use promoted by the European Directive 2010/63/EU. Furthermore, zebrafish exhibits a variety of complex behaviors including, social, learning and anxiety responses, offering a powerful and versatile model system for the study of behavioral neuroscience and related fields [27], [28], [29], [30]
The main objective of this work was to evaluate the toxicity of a cyanobacterial bloom in a lake of an urban park in the city of Aveiro (Portugal) and to understand whether different stages of evolution of a bloom represent different toxicities. Using embryos of zebrafish, a multilevel assessment was performed to evaluate mortality, hatching, development, biochemical (Glutathione S-transferase (GST), Catalase (CAT), Glutathione Peroxidase (GPx), Glutathione Reductase (GR), Cholinesterase (ChE) and Lactate dehydrogenase (LDH)) and behavioral (total distance, slow and rapid movements, and path angles) parameters.
Comments 6: The section “3.1 Identification of Samples” would be better placed in the methodology, extending the text in section 2.1. You could mention that after sample collection, the samples were identified [...].
Response 6: We placed the identification of the samples in the methodology, extending the text of section 2.1 collection of samples, as you suggested, page 3, lines 134-137:
The bloom samples were observed under a light microscopy (Olympus CX31), and the taxonomic study was based on Komárek and co-autors work ([31]), identifying the presence of Microcystis aeruginosa.
Comment 7: Lines 288-291: “The following biomarkers were measured in the larvae exposed to different concentrations of cyanobacterial extract: Glutathione S-transferase (GST), Catalase (CAT), Glutathione Peroxidase (GPx), Glutathione Reductase (GR), Cholinesterase (ChE) and Lactate dehydrogenase (LDH).” This information has already been clearly stated in the methodology, so there is no need to repeat it here.
Response 7: We agree and have removed this information from the results.
Comment 8: I found the lack of a paragraph discussing the limitations of the study to be a significant omission. Particularly, the absence of quantitative measurements of cyanotoxins in the samples should be addressed, along with any other notable limitations.
Response 8: Thank you for pointing this. We apologise that the quantification of the samples was not included in the manuscript by mistake. We have added a section on materials and methods, quantification, page 4, lines 166-176:
2.3. Quantification
To determine the group and concentration of cianotoxins in the bloom and water column samples, the Microcystin-ADDA ELISA Kit (Enzo Life Sciences, ALX-850-319-KI01), an immunoassay for the quantitative detection of microcystins and nodularins in water samples through indirect competitive ELISA, was used according to the manufacturer’s instructions. For each bloom (site 1 and site 2) three dilutions were made: 5x, 10x and 50x. The water column samples from both sites were analyzed with 5x and 10x dillutions. The immunoassay showed us that the samples were composed by microcystins, however, the dilutions made were not sufficient for the environmental samples to fit the standards curve, which means that the concentration of microcystins of the samples were higher than the maximum concentration of the KIT standards (5.0 ppb).
Comment 9: Lines 519-521: “In conclusion, following the pattern of the extracts, filtrates from site 1 presented more toxicity than site 2, confirming the hypothesis that the two sites represent two stages of the bloom life cycle.” Was this the hypothesis of the study? If so, this was not clearly presented earlier in the manuscript. If this was indeed the hypothesis, I recommend making it explicit in the introduction and methodology sections as well.
Response 9: We have added a sentence to clarify, in introduction paragraph describing the aim of the study, page 3, lines 116-123:
The main objective of this work was to evaluate the toxicity of a cyanobacterial bloom in a lake of an urban park in the city of Aveiro (Portugal) and to understand whether different stages of evolution of a bloom represent different toxicities. Using embryos of zebrafish, a multilevel assessment was performed to evaluate mortality, hatching, development, biochemical (Glutathione S-transferase (GST), Catalase (CAT), Glutathione Peroxidase (GPx), Glutathione Reductase (GR), Cholinesterase (ChE) and Lactate dehydrogenase (LDH)) and behavioral (total distance, slow and rapid movements, and path angles) parameters.
Comments 10: Although the journal does not impose strict limitations on the number of references, 107 references for a case study seems excessive. While I appreciate that I did not identify any self-citations (which is positive), there are instances where the authors cite numerous studies to clarify a point that could be sufficiently addressed with 2 or 3 references. I suggest revisiting the references and aiming to reduce the overall number where possible.
Response 10: Thank you for pointing this. We analyzed the manuscript and agreed that we had too many references to support the same sentence/point of view. Therefore, we reviewed the text and removed non-essential references reducing the number to 77.
Round 2
Reviewer 2 Report
Comments and Suggestions for Authors
I am pleased to confirm that the authors have successfully addressed all the suggested revisions. The manuscript now possesses all the qualities and merits that justify its publication. I congratulate the authors on the hard work and dedication invested in this study. Therefore, I recommend accepting the article for publication.